# Early-onset autoimmune vitiligo associated with an enhancer variant haplotype that upregulates class II HLA expression

Ying Jin[1,2], Genevieve H.L. Roberts[1], Tracey M. Ferrara[1], Songtao Ben[1], Nanja van Geel[3], Albert Wolkerstorfer[4], Khaled Ezzedine[5], Janet Siebert[6], Charles P. Neff[7], Brent E. Palmer[7], Stephanie A. Santorico[1,8] & Richard A. Spritz[1,2]

Vitiligo is an autoimmune disease in which melanocyte destruction causes skin depigmentation, with 49 loci known from previous GWAS. Aiming to define vitiligo subtypes, we discovered that age-of-onset is bimodal; one-third of cases have early onset (mean 10.3 years) and two-thirds later onset (mean 34.0 years). In the early-onset subgroup we found novel association with MHC class II region indel rs145954018, and independent association with the principal MHC class II locus from previous GWAS, represented by rs9271597; greatest association was with rs145954018del-rs9271597A haplotype ($P = 2.40 \times 10^{-86}$, OR = 8.10). Both rs145954018 and rs9271597 are located within lymphoid-specific enhancers, and the rs145954018del-rs9271597A haplotype is specifically associated with increased expression of *HLA-DQB1* mRNA and HLA-DQ protein by monocytes and dendritic cells. Thus, for vitiligo, MHC regulatory variation confers extreme risk, more important than *HLA* coding variation. MHC regulatory variation may represent a significant component of genetic risk for other autoimmune diseases.

[1] Human Medical Genetics and Genomics Program, University of Colorado School of Medicine, Aurora 80045 CO, USA. [2] Department of Pediatrics, University of Colorado School of Medicine, Aurora 80045 CO, USA. [3] Department of Dermatology, Ghent University Hospital, Ghent 9000, Belgium. [4] Netherlands Institute for Pigment Disorders, Department of Dermatology, Academic Medical Centre University of Amsterdam, Amsterdam 1100 DD, The Netherlands. [5] Department of Dermatology, Hôpital Henri Mondor, Université Paris-Est Créteil, Créteil 94000, France. [6] CytoAnalytics, Denver 80113 CO, USA. [7] Department of Medicine, University of Colorado School of Medicine, Aurora 80045 CO, USA. [8] Department of Mathematical and Statistical Sciences, University of Colorado, Denver 80204 CO, USA. These authors made equal contributions: Ying Jin, Genevieve H.L. Roberts Correspondence and requests for materials should be addressed to R.A.S. (email: richard.spritz@ucdenver.edu)

Vitiligo is an autoimmune disease in which destruction of skin melanocytes results in patches of white skin and hair[1]. In three previous genome-wide association studies (GWAS), we identified 49 genetic loci associated with vitiligo susceptibility[2–6], most of which harbor genes involved in regulation of immune cells, apoptosis, and melanocyte function. These fit together in a general model of melanocyte autoimmune pathogenesis[7].

Vitiligo is frequently associated with other autoimmune diseases, particularly autoimmune thyroid disease, type 1 diabetes, pernicious anemia, rheumatoid arthritis, systemic lupus erythematosus, and Addison disease[8], and a number of vitiligo susceptibility loci are shared with these other diseases[7]. Chief among these is the Major Histocompatibility Complex (MHC), with vitiligo having independent genetic associations in both the MHC class I and class II regions[2,4,5]. However, unlike for many other autoimmune diseases, for vitiligo principal MHC associations localize to intergenic non-coding regions[9,10], rather than coding variants that alter HLA protein structure and thereby affect binding and presentation of peptide antigens.

In the present study, we investigate clinical variation among vitiligo cases in an attempt to define vitiligo subgroups and to then explore differential underlying genetic basis. We began by characterizing secondary vitiligo phenotypes, starting with age-of-onset and association with other autoimmune diseases. Unexpectedly, we find that vitiligo age-of-onset is bimodal, consisting of early-onset and late-onset subgroups. To investigate genetic differences between these two subgroups, we then categorized vitiligo cases as early-onset or late-onset and carried out stratified GWAS of each subgroup separately. Specifically in the early-onset subgroup, we identify a novel, very strong association with rs145954018, an insertion-deletion (indel) polymorphism in the MHC class II region. In both the early- and late-onset subgroups we also observe independent association with a separate MHC class II locus found by our previous vitiligo GWAS, represented by rs9271597. Extreme vitiligo risk and early disease onset are associated with the rs145954018del-rs9271597A haplotype that includes the risk alleles of both variants; coding variation within

the classical HLA alleles on this haplotype does not independently contribute to vitiligo risk. Surprisingly, we also observe lower frequency of other, concomitant autoimmune diseases in early-onset vitiligo cases than in late-onset cases. This may be explained by the protective effects on these autoimmune diseases of HLA-DRB1*13:01, which typically occurs on the background of the early-onset rs145954018del-rs9271597A haplotype.

Both rs145954018 and rs9271597 are located within different lymphoid-specific enhancers predicted by ENCODE[11]. Accordingly, we examined the effect of each variant on expression of neighboring HLA genes. We find that rs145954018del and the early-onset rs145954018del-rs9271597A haplotype are specifically associated with significantly elevated expression of HLA-DQ mRNA and protein by professional antigen presenting cells including peripheral blood monocytes and dendritic cells. Thus, for vitiligo, extreme genetic risk and early disease onset are genetically associated with a MHC class II haplotype that is associated with increased HLA-DQ expression, rather than with specific HLA alleles that produce structurally different HLA proteins.

## Results

**Vitiligo consists of early-onset and late-onset subgroups.**
Among the total 4523 vitiligo cases of European ancestry in our three previous GWAS and replication cohorts, the overall mean age-of-onset was 25.9 years, SD 16.6 (Fig. 1), with no significant difference between males and females; males mean 26.5 years, SD 16.8 and females mean 25.6 years, SD 16.5 (Supplementary Figure 1), and the mean age-of-onset was similar in each of the four constituent vitiligo case cohorts (GWAS1, mean 24.0 years, SD 16.4; GWAS2, mean 27.5 years, SD 16.9; GWAS3, mean 27.3 years, SD 16.4; replication cohort, mean 26.3 years, SD 16.7). However, in all four case cohorts the age-of-onset distribution appeared non-normal, in both males and females (Supplementary Figure 2). Statistical goodness of fit analysis supported a bimodal age-of-onset distribution composed of two overlapping normally distributed subgroups (Fig. 1, Supplementary Table 1, and

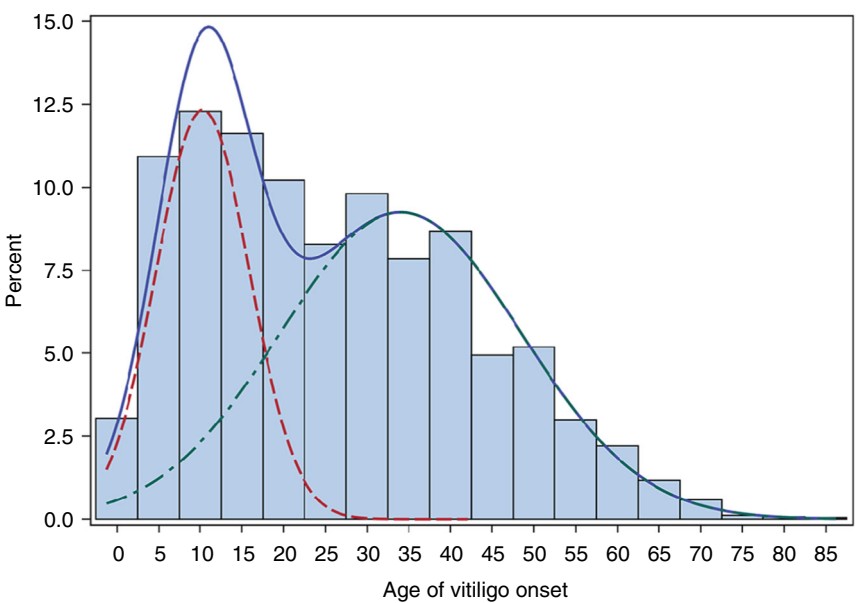

**Fig. 1** Vitiligo age-of-onset is bimodal. The distribution of vitiligo age-of-onset and resultant finite mixture model is shown for the total 4523 cases from our three previous vitiligo GWAS and replication study[2,4,5]. The blue line shows the full distribution, the red line the early-onset distribution (mean 10.3, SD 5.6 years; 38.4%), and the green line the late-onset distribution (mean 34.0, SD 14.5 years; 61.6%). The overall and sex-specific distributions, resultant finite mixture models, and overall and sex-specific mean ages-of-onset were very similar among the four constituent case cohorts. See also Supplementary Figures 1, 2, and 3. Source data are provided as a Source Data file

Supplementary Figure 3): an early-onset subgroup (mean 10.3 years, SD 5.6; 38.4%) and a late-onset subgroup (mean 34.0 years, SD 14.5; 61.6%), with similar proportions of early-onset and late-onset cases in the four constituent case cohorts (37.5:62.5%; 37.2:62.8%; 41.4:58.6%; 37.6:62.4%; Supplementary Figure 3).

**Early-onset vitiligo associated with MHC indel rs145954018**. To explore differential genetic underpinnings of early-onset vs. late-onset vitiligo, we assigned cases from each GWAS cohort to these subgroups, including only cases that could be classified with >80% probability of belonging to one or the other subgroup. Cases were then genetically matched with controls from the corresponding GWAS[12]. After subgroup assignment and control matching, the combined early-onset subgroup contained 704 cases and 9,031 controls and the combined late-onset subgroup contained 1,467 cases and 19,156 controls. We then carried out separate GWAS for each subgroup and tested effect size differences for variants that showed genome-wide significant association ($P_{CMH} < 5.0 \times 10^{-8}$) with vitiligo in either or both subgroups.

A total 5,493 MHC and 771 non-MHC variants showed genome-wide significant association in one or both age-of-onset subgroups (Supplementary Data File 1). As shown in Table 1 and Fig. 2, in the early-onset subgroup greatest association was with two-base indel variant (TG/del) rs145954018 ($P_{logistic} = 3.47 \times 10^{-54}$, OR = 4.62, 95% CI 3.81–5.61), located in the MHC class II region; rs145954018 did not achieve genome-wide significance in the late-onset subgroup ($P_{logistic} = 4.50 \times 10^{-5}$, OR = 1.47, 95% CI 1.22–1.75). In the late-onset subgroup, greatest association was with the overall most significant variant in our previous vitiligo GWAS[5], MHC class II SNP (T/A) rs9271597 ($P_{logistic} = 3.51 \times 10^{-34}$, OR = 1.63, 95% CI 1.51–1.77), which also showed strong association in the early-onset case subgroup ($P_{logistic} = 2.31 \times 10^{-40}$, OR = 2.23, 95% CI 1.98–2.51) (Table 1 and Fig. 2). Thus, rs145954018 is specifically associated with early-onset vitiligo whereas rs9271597 is generically associated with vitiligo risk.

Stepwise logistic regression analysis indicated that rs145954018 was the only variant representing the early-onset association signal, whereas rs9271597 as well as rs9271600 and rs9271601, all represent the generic association signal; these three SNPs are located only 45 bases apart and are perfectly correlated ($r^2 = 1.0$) in the European-ancestry population[9]. No other variants in the MHC showed significantly different effect sizes in the early-versus late-onset vitiligo subgroups when the effects of rs145954018 and rs9271597 were accounted for in stepwise logistic regression analysis (Supplementary Data File 2). Additionally, no variants elsewhere in the genome (Supplementary Data File 2), including the 49 other previous vitiligo susceptibility loci (Supplementary Table 2), showed significantly different effect sizes comparing the early- and late-onset vitiligo subgroups.

As a replication test, we stratified our previous GWAS replication case cohort into early-onset and late-onset subgroups as above, randomly assigned controls to each subgroup, and genotyped rs145954018 and rs9271597. As shown in Table 1, rs145954018 showed far greater association and stronger effect in the early-onset replication subgroup ($P_{logistic} = 1.57 \times 10^{-30}$, OR = 5.48 95% CI 4.10–7.32) than in the late-onset replication subgroup ($P_{logistic} = 0.002$, OR = 1.67, 95% CI 1.21–2.31). SNP rs9271597 again was associated in both the early-onset subgroup ($P_{logistic} = 5.54 \times 10^{-23}$, OR = 2.40, 95% CI: 2.02–2.85) and the late-onset subgroup ($P_{logistic} = 2.28 \times 10^{-10}$, OR = 1.57, 95% CI: 1.37–1.81). These results confirm that rs145954018 is specifically associated with early-onset vitiligo, while rs9271597 is associated in both subgroups.

**Early vitiligo onset and rs145954018del-rs9271597A haplotype**. rs145954018 and rs9271597 are in almost complete LD (D' = 0.98, $r^2 = 0.04$). The relatively uncommon rs145954018del risk allele (MAF = 3.5% in our GWAS controls) almost always occurs on the background of the relatively common risk allele of generic SNP rs9271597 (MAF = 42.7%), together constituting a haplotype that carries the risk alleles of both variants, rs145954018del-rs9271597A. To determine whether rs145954018 and rs9271597 are independently associated with vitiligo susceptibility, we carried out multiple logistic regression analysis including both variants in the same model (Supplementary Table 3). Adjusting for rs9271597, rs145954018 was significantly associated with vitiligo in the early-onset subgroup ($P_{conditional} = 1.01 \times 10^{-31}$, OR = 3.35, 95% CI: 2.74–4.10), with a significantly ($P_Z = 2.27 \times 10^{-14}$) higher OR than in the late-onset subgroup ($P_{conditional} = 0.16$, OR = 1.15, 95% CI: 0.95–1.38). Adjusting for rs145954018, the generic SNP, rs9271597, was significantly associated with vitiligo in both the early-onset ($P_{conditional} = 2.99 \times 10^{-24}$, OR = 1.91, 95% CI 1.68–2.16) and late-onset ($P_{conditional} = 2.04 \times 10^{-31}$, OR = 1.62, 95% CI: 1.49–1.75) subgroups, with a marginally ($P_Z = 0.03$) higher OR in the early-onset subgroup. The replication study data yielded similar results (Supplementary Table 3). These results show that both rs145954018 and rs9271597 are independently associated with vitiligo in the early-onset case subgroup.

To visualize the relationship between the rs145954018del-rs9271597A haplotype and vitiligo age-of-onset, we plotted haplotype frequency versus vitiligo age-of-onset. As shown in Fig. 3, the frequency of the rs145954018del-rs9271597A haplotype was strikingly non-linear, exhibiting dramatic elevation in vitiligo cases with age-of-onset from about 5 to 9 years

### Table 1 Principal MHC class II variants in the early- and late-onset subgroups

| Variant | Chr 6 Position (Build 37) | EA/OA | Early-onset | | | Late-onset | | | Effect size difference[a] | |
| | | | P value | OR | SE | P value | OR | SE | Z score | P value |
|---|---|---|---|---|---|---|---|---|---|---|
| *Combined GWAS*[a] | | | | | | | | | | |
| rs145954018 | 32440321 | D/I | $3.47 \times 10^{-54}$ | 4.62 | 0.10 | $4.50 \times 10^{-5}$ | 1.47 | 0.09 | 8.42 | $3.75 \times 10^{-17}$ |
| rs9271597 | 32591291 | A/T | $2.31 \times 10^{-40}$ | 2.23 | 0.06 | $3.51 \times 10^{-34}$ | 1.63 | 0.04 | 4.31 | $1.64 \times 10^{-5}$ |
| *Replication study*[b] | | | | | | | | | | |
| rs145954018 | 32440321 | D/I | $1.57 \times 10^{-30}$ | 5.48 | 0.15 | 0.002 | 1.67 | 0.17 | 5.32 | $1.04 \times 10^{-7}$ |
| rs9271597 | 32591291 | A/T | $5.54 \times 10^{-23}$ | 2.40 | 0.09 | $2.28 \times 10^{-10}$ | 1.57 | 0.07 | 3.70 | $2.16 \times 10^{-4}$ |

Chr chromosome, EA effect allele, OA other allele, SE standard error of the logarithm of the OR
[a]Obtained by logistic regression analysis including significant eigenvectors from each GWAS as covariates
[b]Obtained by logistic regression analysis
[c]A Z test[32] was used to test effect size difference, with Z calculated as the difference between the logarithms of the ORs, divided by the square root of the sum of the variances of the logarithms of the ORs, where Z follows a normal distribution

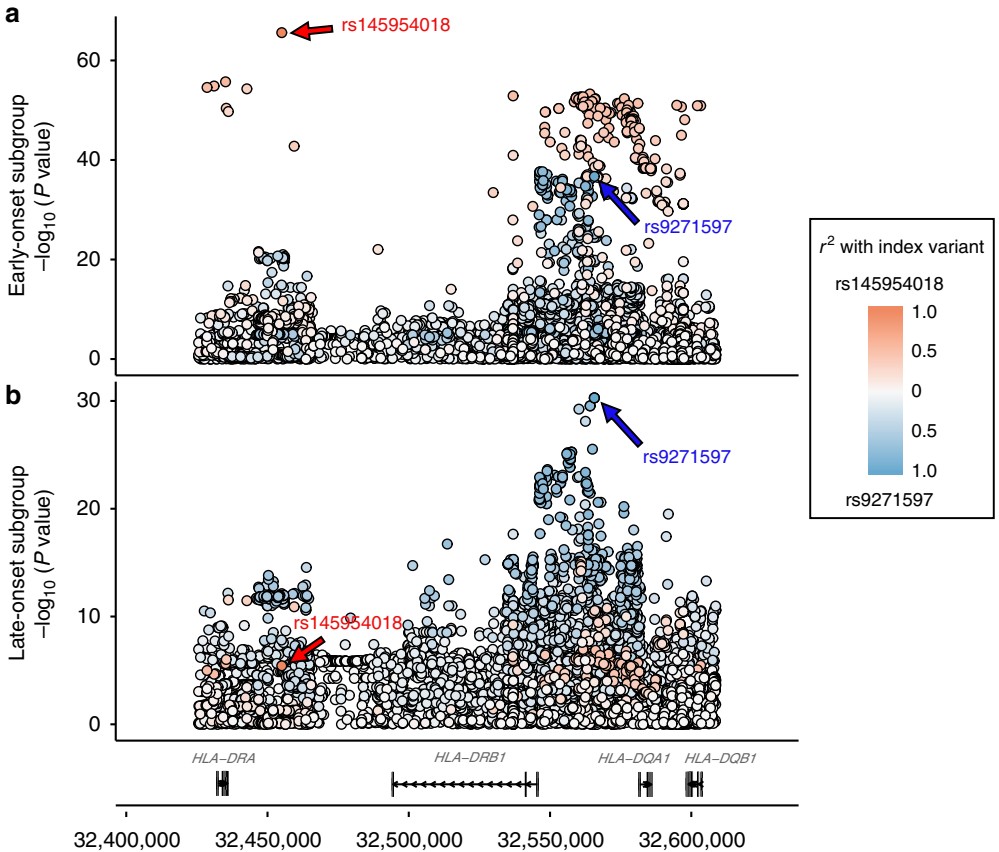

**Fig. 2** MHC class II associations in early- and late-onset vitiligo subgroups. $-\log_{10} P_{CMH}$ values are shown for variants tested across the *HLA-DRA* through *HLA-DQB1* segment of the MHC class II region in the separate (**a**) early-onset and (**b**) late-onset case-control cohorts from the combined GWAS. Red indicates higher $r^2$ with rs145954018, blue indicates higher $r^2$ with rs9271597, and white indicates low correlation with either rs145954018 or rs9271597. Arrows show rs145954018 and rs9271597. Genes and nucleotide positions are from GRCh37/hg19. Note that $-\log_{10} P_{CMH}$ value scales differ for early-onset and late-onset subgroups. Source data are provided as a Source Data file

(OR = 11.01, 95% CI: 8.75–13.87, $P_{logistic} = 4.49 \times 10^{-92}$), with frequency of 18.5% in this group of cases versus about 3.5% in controls. These results show that the rs145954018del-rs9271597A haplotype is associated with remarkably high risk in children with vitiligo onset at ages 5–9 years. Surprisingly, the mean age of vitiligo onset was very similar in vitiligo cases carrying one (mean 19.8 years, SD 18.9) vs. two (mean 20.8 years, SD 15.1) copies of the high-risk rs145954018del-rs9271597A haplotype, compared to cases carrying only the reference rs145954018TG-rs9271597T haplotype (mean 29.3 years, SD 16.8). This suggested that the rs145954018del-rs9271597A haplotype might exert a dominant effect, accelerating vitiligo age-of-onset by approximately nine years. Comparison of additive and dominant logistic regression models indicated that the rs145954018del-rs9271597A haplotype indeed acts as a complete dominant, dramatically increasing vitiligo risk, with estimated OR 8.10 ($P = 2.40 \times 10^{-86}$) (Supplementary Table 4).

**Early vitiligo onset not driven by *HLA* coding variation.** Imputation of MHC class II classical *HLA* alleles[13] in our GWAS data, analysis of classical *HLA* allele sequences in 20 genotyped vitiligo cases (Supplementary Table 5), and alignment with European-American MHC class II haplotype reference standards[14] showed that the rs145954018del risk allele resides on an extended MHC class II haplotype, rs145954018del–*HLA-DRB1\*13:01*–*HLA-DRB3\*01:01*–rs9271597A–*HLA-DQA1\*01:03*–*HLA-DQB1\*06:03*. To dissect the relative contributions of rs149554018, rs9271597, and the classical *HLA* alleles to

vitiligo risk, we compared logistic regression models with haplotypes defined by different combinations of rs145954018, *HLA-DRB1\*13:01*, rs9271597, *HLA-DQA1\*01:03*, and *HLA-DQB1\*06:03*. The best fit model included only rs145954018 and rs9271597 ($P_{logistic} = 3.85 \times 10^{-83}$); inclusion of any of the classical *HLA* alleles failed to improve the model (Supplementary Table 6). Together, these findings indicate that extreme vitiligo risk and early onset is principally associated with the rs14595401del-rs9271597A haplotype; the classical *HLA* alleles on the extended high-risk haplotype do not appear to show additional independent effects on vitiligo risk.

**rs145954018, rs9271597, and vitiligo heritability.** Both rs145954018 and rs9271597 showed larger effects in the early-onset subgroup than the late-onset subgroup, so we compared their contributions to vitiligo heritability (the fraction of risk attributable to additive genetic variation; $h^2$) in the two subgroups. In the early-onset subgroup, total vitiligo $h^2 = 0.55$, of which rs145954018 accounts for 7.5%, rs9271597 accounts for 6.8%, and the two together 11.4% of the total. In the late-onset subgroup, total vitiligo $h^2 = 0.40$, of which rs145954018 only accounts for 0.4%, rs9271597 for 3.6%, and the two together 3.7%. The other 49 vitiligo susceptibility loci identified by GWAS altogether account for very similar fractions of vitiligo $h^2$ in the early-onset (13.3%) and late-onset (15.0%) subgroups. The fraction of vitiligo $h^2$ explained by rs145954018 is nearly 20-fold higher in the early-onset subgroup than in the late-onset subgroup, hence explaining a large proportion of the difference in heritability between the early- and late-onset subgroups.

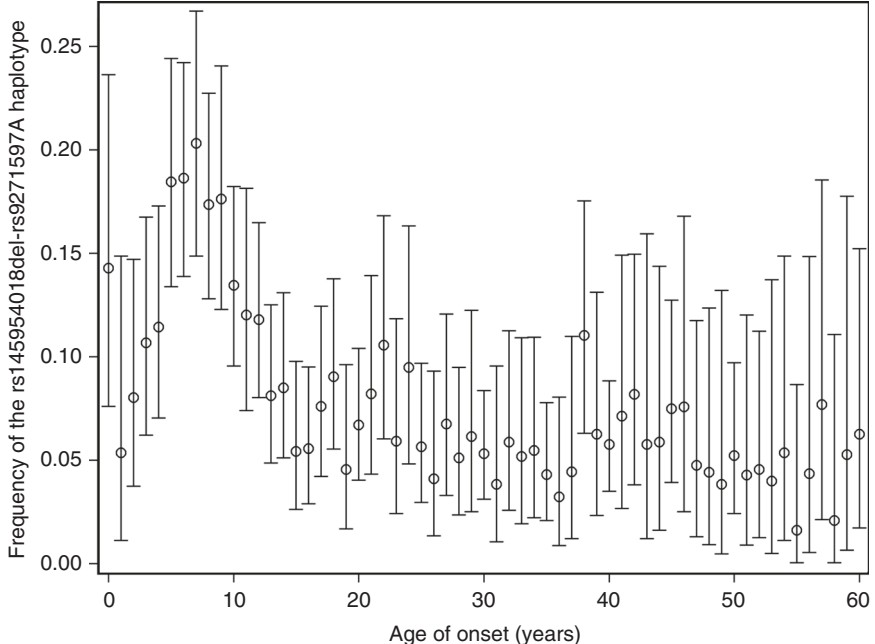

**Fig. 3** rs1459540del-rs9271597A haplotype frequency across vitiligo age-of-onset. Frequency of the rs145954018del-rs9271597A high-risk haplotype in 4,254 vitiligo cases is shown in 1-year age-of-onset intervals. Circles indicate means and bars 95% confidence intervals, calculated using the binom.test function in R. Note that only cases with age-of-onset < 60 years were used to make the figure, as beyond age-of-onset 60 years the number of cases in each 1-year interval is too small to give reliable estimates of means and 95% confidence intervals. Source data are provided as a Source Data file

**rs145954018 increases *HLA-DQB1* mRNA and HLA-DQ protein**. The rs145954018 indel is located within an immune-specific transcriptional enhancer (ENSEMBL Human Regulatory Feature ENSR00000195683), between *HLA-DRA* and *HLA-DQA1* (Fig. 4). Likewise, rs9271597 is located within another immune-specific enhancer (ENSEMBL Human Regulatory Feature ENSR00000195696), between *HLA-DRB1* and *HLA-DQA1*, which we previously found is associated with surface expression of HLA-DQ and HLA-DR proteins on peripheral blood monocytes[9]. We therefore hypothesized that rs145954018 and rs9271597 might both affect expression of HLA class II proteins, either independently or synergistically. To test this, we assayed surface expression of HLA-DQ and HLA-DR proteins on sorted peripheral blood monocytes, dendritic cells, and B cells from 46 healthy subjects selected on the basis of rs145954018 and rs9271597 genotypes and inferred haplotypes. As shown in Fig. 5, the rs14595401del-rs9271596A haplotype and rs145954018del allele (for which the results were identical because of LD) were both associated with increased expression of HLA-DQ protein on monocytes ($P_{linear} = 4.84 \times 10^{-4}$; Fig. 5a) and dendritic cells ($P_{linear} = 0.005$; Fig. 5b), but were not associated with expression of HLA-DR protein in either cell type ($P_{linear} = 0.96$ and $P_{linear} = 0.81$, respectively). Stepwise regression analysis showed that, after adjusting for rs145954018, rs9271597A was marginally associated with increased expression of HLA-DR ($P_{conditional} = 0.04$) on monocytes, but was not associated with increased expression of HLA-DQ on any of the cell types studied.

To assess whether the rs145954018del-rs9271597A haplotype specifically affects expression of the *HLA* class II genes on the same haplotype (in *cis*), we carried out allele-specific RNA-seq analysis of peripheral blood RNA from three healthy subjects from the protein study. Subject 1 carried two low-risk rs145954018TG-rs9271597T haplotypes, while subjects 2 and 3 each carried one low-risk haplotype and one high-risk rs145954018del-rs9271597A haplotype. As shown in Table 2, subject 1 expressed similar amounts of *HLA-DRB1*, *HLA-DQA1*, and *HLA-DQB1* mRNA from each chromosome. However, the

two subjects heterozygous for the high-risk rs145954018del-rs9271597A haplotype expressed about 2.6- and 2.1-fold more total *HLA-DQB1* mRNA, respectively. Furthermore, both subjects expressed proportionately more transcripts from the *HLA-DQB1\*06:03* allele in *cis* on the high-risk rs145954018del-rs9271597A haplotype, than from the *HLA-DQB1* allele on the low-risk haplotype. These results indicate that the high-risk rs145954018del-rs9271597A haplotype is specifically associated with increased expression of the *HLA-DQB1\*06:03* allele on the extended haplotype.

**Early vitiligo onset and risk of other autoimmune diseases**. Vitiligo is epidemiologically associated with increased risk of autoimmune thyroid disease, type 1 diabetes, pernicious anemia, rheumatoid arthritis, systemic lupus erythematosus, and Addison's disease[8,15]. We considered that greater genetic risk for vitiligo in the early-onset subgroup might result in greater frequency of concomitant occurrence of these other autoimmune diseases in early-onset cases than in late-onset cases. Surprisingly, we observed the opposite: the frequency of other vitiligo-associated autoimmune diseases was significantly lower ($P_{Fisher} = 4.05 \times 10^{-4}$) in early-onset cases (17.0%) than in late-onset cases (22.4%, Supplementary Table 7).

The extended rs145954018del-rs9271597A haplotype is in almost complete LD with *HLA-DRB1\*13:01* ($D' = 0.98$, $r^2 = 0.53$). *HLA-DRB1\*13:01* is associated with relative protection from systemic lupus erythematosus, rheumatoid arthritis[16], and type 1 diabetes[17] in European-derived populations. Thus, while rs145954018del is associated with extreme vitiligo risk and early disease onset, *HLA-DRB1\*13:01* might simultaneously confer relative protection from other autoimmune diseases, and thus might account for the reduced frequency of concomitant autoimmunity in early-onset vitiligo cases. To test this, we analyzed association between non-vitiligo autoimmunity and *HLA-DRB1\*13:01* and rs145954018. Individually, both *HLA-DRB1\*13:01* ($P_{logistic} = 0.004$, OR = 0.71, 95% CI 0.56-0.90) and

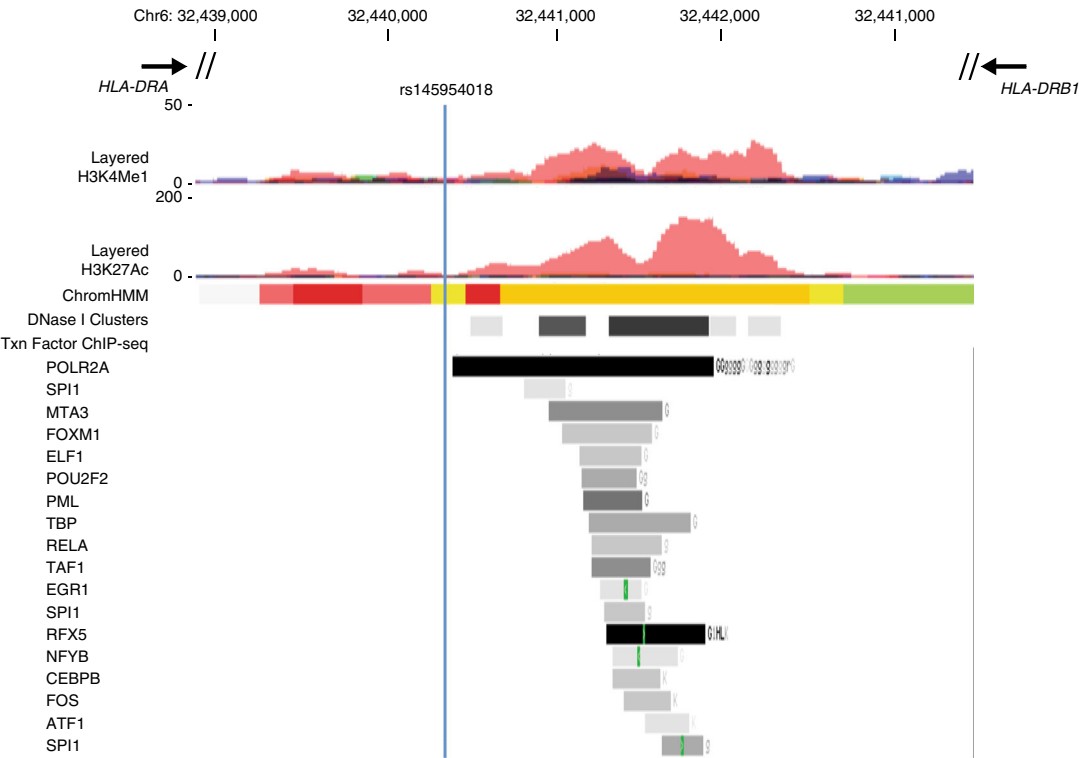

**Fig. 4** rs145954018 is located within transcriptional enhancer ENSR00000195683. The blue vertical line indicates the position of the rs145954018 indel variant. Layered H3K27Ac and H3K4Me1 marks, hidden Markov model chromatin state segmentation (ChromHMM), DNase I hypersensitive site cluster (DNase I Clusters), and transcription factor chromatin immunoprecipitation sequencing (abbreviated Txn Factor ChIP-seq) data are from ENCODE[11,36]. For layered H3K27Ac and H3K4Me1 marks, data are shown for the seven cell lines studied by ENCODE. For ChromHMM, red indicates active promoter, orange indicates strong enhancer, and yellow indicates weak/poised enhancer; data shown are for GM12878 B lymphoblastoid cells. For DNase clusters, darkness indicates relative signal strength in 125 cell types from ENCODE (V3). For Txn factor ChIP-seq, darkness indicates relative signal strength of aggregate binding of 161 transcription factors, and green bars indicate ENCODE Factorbook[37] canonical motifs for specific transcription factors

rs145954018del ($P_{logistic} = 0.04$, OR = 0.74, 95% CI: 0.55–0.98) showed significant protective association against non-vitiligo autoimmune diseases. However, after adjusting for the effect of rs145954018, *HLA-DRB1\*13:01* was marginally protective against non-vitiligo autoimmunity ($P_{conditional} = 0.05$, OR = 0.69, 95% CI: 0.48–1.00), whereas adjusting for the effect of *HLA-DRB1\*13:01*, rs145954018del showed no association ($P = 0.88$, OR = 1.04, 95% CI: 0.66–1.63) with non-vitiligo autoimmunity. These results suggest that reduced occurrence of concomitant autoimmunity in early-onset vitiligo cases is primarily associated with the *HLA-DRB1\*13:01* allele rather than with rs145954018del.

## Discussion

We have found that autoimmune vitiligo consists of two clinical subgroups: early-onset (mean onset 10.3 years) and late-onset (mean 34.0 years). At least two other autoimmune diseases that are epidemiologically associated with vitiligo[8,15], type 1 diabetes and rheumatoid arthritis, likewise have early-onset and late-onset forms[18,19]. For all three of these diseases, the magnitude of associations and effect sizes of MHC variants are greater in early-onset cases than late-onset cases[20,21]. Thus, for all three diseases the MHC appears to play a special role in mediating age-of-onset.

Early vitiligo onset is specifically associated with the MHC class II region indel rs145954018, which contributes substantially to the heritability difference between the early-onset and late-onset subgroups. The high-risk deletion allele is uncommon, with MAF 3 to 4% in most human populations [https://www.ncbi.nlm.nih.gov/projects/SNP/snp_ref.cgi?rs=145954018]. Its frequency is much lower, with MAF about 0.5%, in east Asians, a population

in which the prevalence of vitiligo is likewise much lower than in others[22–24]. In contrast, the MAF of the rs145954018del risk allele is 15% among early-onset vitiligo cases and is 19% among cases with disease onset from ages 5–9 years. The rs145954018del risk allele is in almost complete LD with the most significant MHC class II locus found by our previous GWAS, rs9271597, which is generically associated with both early- and late-onset vitiligo. The rs145954018del-rs9271597A haplotype carrying the risk alleles of both variants confers a remarkable 8-fold elevation of vitiligo risk and dominantly accelerates vitiligo onset by about 9 years. Nevertheless, even among early-onset vitiligo cases only 28% carry the high-risk rs145954018del-rs9271597A haplotype, indicating that other factors also contribute to early vitiligo onset.

The high-risk rs145954018del-rs9271597A haplotype resides on the extended MHC class II haplotype rs145954018del-*HLA-DRB1\*13:01-HLA-DRB3\*01:01*-rs9271597A-*HLA-DQA1\*01:03-HLA-DQB1\*06:03*. The main effects of the extended haplotype on vitiligo risk and age-of-onset are attributable to rs145954018, and to a lesser extent rs9271597; the classical *HLA* alleles in LD on the extended haplotype do not appear to show additional independent effects. Nevertheless, these classical *HLA* alleles might exert other effects, and protection by *HLA-DRB1\*13:01* likely accounts for the unexpectedly lower frequency of other autoimmune diseases in early-onset vitiligo cases than in late-onset cases.

rs145954018 and rs9271597 are both located within lymphoid-specific transcriptional enhancers. The high-risk rs145954018del allele and the rs145954018del-rs9271597A haplotype are specifically associated with increased expression of HLA-DQ protein on peripheral monocytes and dendritic cells. Taken together, our

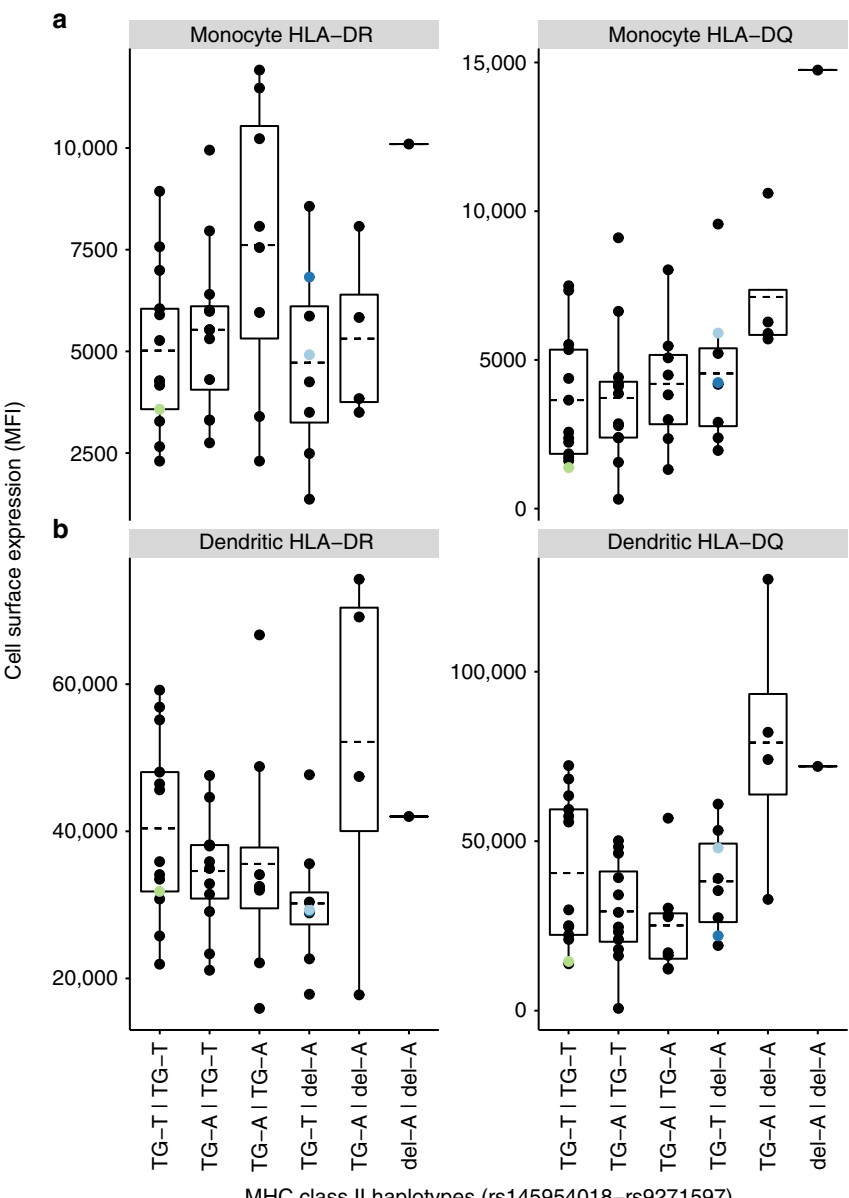

**Fig. 5** HLA-DQ and -DR protein expression by rs145954018-rs9271597 genotype. **a** Monocytes (CD3−, CD11b + , CD11c−, CD14 + , CD19−); **b** Dendritic cells (CD3−, CD11b−, CD11c + , CD14−, CD19−). HLA-DQ and HLA-DR cell surface expression is calculated as mean fluorescence intensity (MFI). Phased rs145954018-rs9271597 genotypes are indicated (GT-T | GT-T, $n = 13$; GT-T | GT-A, $n = 12$; GT-A | GT-A, $n = 8$; GT-T | del-A, $n = 8$; GT-A | del-A, $n = 4$; del-A | del-A, $n = 1$). Circles indicate individual subjects (one measurement per subject), dashed lines denote means, boxes denote first through third quantiles. Each vertical bar extends from the box to the largest or smallest value no farther than 1.5 times the inter-quartile range. Data beyond the vertical bars are considered outliers and are plotted individually. Colored circles represent subjects for whom whole-blood RNA-seq data are presented in Table 2 (green, subject 1; dark blue, subject 2; light blue, subject 3). Source data are provided as a Source Data file

findings are consistent with a model in which the effects of rs145954018del and rs9271597A are synergistic, the rs145954018del-rs9271597A haplotype increasing expression of *HLA-DQB1* mRNA thereby increasing expression of HLA-DQ protein on the surface of professional antigen presenting cells. Increased surface presentation of triggering antigens by HLA-DQ molecules may increase the probability of autoreactive T-cell activation, and thus may both increase overall vitiligo risk and accelerate vitiligo onset. This process is presumably initiated by environmental insult, though it is unknown what triggers might especially pertain to the 5–9 year age group, which appears to be particularly relevant to vitiligo associated with the rs145954018del-rs9271597A haplotype.

Extreme vitiligo risk and early disease onset is thus specifically associated with a haplotype of MHC class II regulatory variants that quantitatively affect HLA protein expression. For other autoimmune diseases, most studies have focused on the role of coding variations that encode structurally different HLA proteins. Regulatory variation in the MHC remains largely unexplored, and may play a generally unrecognized yet significant role in the pathogenesis of many autoimmune diseases.

## Methods

**Study subjects**. GWAS cases were from our three previous vitiligo GWAS1 (1,381 cases)[2], GWAS2 (413) cases[4], and GWAS3 (1,059 cases)[5], and GWAS control data sets were obtained from the database of Genotypes and Phenotypes (dbGaP).

**Table 2 Allele-specific expression of *HLA* class II genes in whole blood cell mRNA**

| Subject[a] | Subject 1 | Subject 2 | Subject 3 |
|---|---|---|---|
| rs145954018-rs9271597 phased haplotypes | GT-T \| GT-T | del-A \| GT-T | del-A \| GT-T |
| *HLA-DRB1-HLA-DQA1-HLA-DQB1* phased haplotypes | *01:01-*01:01-*05:01 | *13:01-*01:03-*06:03 | *13:01-*01:03-*06:03 |
| | *08:02-*04:01-*04:02 | *03:01-*05:01-*02:01 | *12:01-*05:01-*03:09 |
| *HLA-DRA* RNA Total (RPKM) | 54.8 | 40.0 | 51.2 |
| *HLA-DRB1* RNA Total (RPKM) | 66.4 | 93.5 | 70.4 |
| Allele 1 | *01:01 43% | *13:01 43% | *13:01 58% |
| Allele 2 | *08:02 57% | *03:01 57% | *12:01 42% |
| *HLA-DQA1* RNA Total (RPKM) | 13.1 | 11.0 | 12.3 |
| Allele 1 | *01:01 36% | *01:03 69% | *01:03 79% |
| Allele 2 | *04:01 64% | *05:01 31% | *05:01 21% |
| *HLA-DQB1* RNA Total (RPKM) | 16.5 | 42.8 | 34.2 |
| Allele 1 | *05:01 55% | *06:03 84% | *06:03 69% |
| Allele 2 | *0402 45% | *02:01 16% | *03:09 31% |

For *HLA-DRB1*, *HLA-DQA1*, and *HLA-DQB1* RPKM and the percentage of reads mapping to the most likely *HLA* classical alleles from seq2HLA software[35] are shown, reflecting allele-specific expression.
*HLA-DRA* is non-polymorphic; therefore, allele-specific expression cannot be ascertained.
*RPKM* Reads Per Kilobase of transcript, per Million mapped reads (normalized RNA expression)
[a]The data are related to Fig. 5

Control datasets were phs000092.v1.p1, phs000125.v1.p1, phs000138.v2.p1, phs000142.v1.p1, phs000168.v1.p1, phs000169.v1.p1, phs000206.v3.p2, phs000237.v1.p1, phs000346.v1.p1, and phs000439.v1.p1 for GWAS1; phs000203.v1.p1, and phs000289.v2.p1 for GWAS2; phs000196.v2.p1, phs000303.v1.p1, phs000304.v1.p1, phs000368.v1.p1, phs000381.v1.p1, phs000387.v1.p1, phs000389.v1.p1, phs000395.v1.p1 phs000408.v1.p1, phs000421.v1.p1, phs000494.v1.p1, and phs000524.v1.p1 for GWAS3). Control datasets were matched to each of the three GWAS case datasets based on platforms used for genotyping. The replication study included 1,743 cases and 2,182 unaffected controls from our previous independent replication study[5]. All subjects were unrelated and of self-described non-Hispanic/Latino European-derived white ancestry from North America and Europe. All cases met diagnostic criteria for generalized vitiligo[1]. We obtained written informed consent from all study participants, and the study was approved by the institutional review board at each participating center, with overall oversight provided by the Colorado Multiple Institutional Review Board (COMIRB).

**Age-of-onset analyses.** All cases provided self-reported vitiligo age-of-onset. To define the mixture of age-of-onset distributions in the GWAS and replication case cohorts, we performed goodness of fit analyses using the finite mixture model (FMM) procedure in SAS version 9.4 [https://www.sas.com]. We set component distributions as normal, the maximum number of mixture components as seven, fit the FMMs using the maximum likelihood method, and chose the best fit model using the Bayesian information criterion (BIC).

**Genotyping, quality control and imputation.** Genome-wide genotyping, quality control procedures, and genome-wide imputation were described previously[2,4,5]. To increase genetic resolution in the MHC, we used the Sanger Imputation Service [https://imputation.sanger.ac.uk/] to impute SNP genotypes using the Haplotype Reference Consortium reference panel (release 1.1) [http://www.haplotype-reference-consortium.org/]. We used Eagle2[25] to pre-phase genotypes to produce best-guess haplotypes and then performed imputation using PBWT[26]. SNPs with imputation INFO > 0.5, minor allele frequency ≥ 0.001 from the three GWAS combined, and without significant ($P < 1 \times 10^{-5}$) deviation from Hardy–Weinberg equilibrium were retained and combined with variants previously genotyped or imputed with IMPUTE2[27] from the previous GWAS;[5] for SNPs imputed by both IMPUTE2 and PBWT, the PBWT imputed genotypes were retained. We imputed classical *HLA* alleles and amino acid polymorphisms for the three previous GWAS using SNP2HLA and the Type 1 Diabetes Genetics Consortium reference panel[13]. In total, 53,976 MHC variants and MHC classical alleles were tested in the final analysis.

In the replication study, rs145954018 and rs9271597 were genotyped using the Applied Biosystems SNaPshot Multiplex System. Both variants had genotype call rates > 95% and were in Hardy–Weinberg equilibrium ($P > 0.05$).

**Homogenous case-control clusters and eigenvectors.** To control for population stratification in the GWAS data, we derived homogeneous case-control clusters and significant eigenvectors for each GWAS cohort separately using GemTools[12] with the default parameter settings. For each GWAS cohort, we used genotyped SNPs with genotype missing rate < 0.1%, MAF > 0.01, and Hardy–Weinberg $P$ value > 0.005, and chose tag SNPs with $r^2 < 0.01$ using PLINK[28] version 1.9. The number of tag SNPs used for GWAS1, GWAS2, and GWAS3 were 13,459, 12,140, and 13,010, respectively. Significant eigenvectors for genetic ancestry were derived from the normalized graph Laplacian, with the number of significant eigenvectors estimated based on the eigengap heuristic and hypothesis testing[29]. Homogeneous case-control clusters were derived by k-means clustering using Ward's algorithm[30],

using a matrix including the significant eigenvectors, normalized using the NJW algorithm[31].

**Early-onset and late-onset subgroup stratification.** To stratify cases from each GWAS into early-onset and late-onset vitiligo subgroups, we included only cases with > 80% posterior probability of belonging to either the early-onset (GWAS1, 1–10 years; GWAS2 1–13; GWAS3 0–14) or late-onset (GWAS1, 19–84 years; GWAS2, 22–72; GWAS3, 21–81) case subgroup based on the best fit FMM for each GWAS. Posterior probabilities were derived from the FMM procedure in SAS version 9.4. All cases assigned to either the early- or late-onset subgroup were assigned to genetically homogeneous clusters with controls from the corresponding cohort using GemTools[12]. We then combined the three early-onset case-control subgroups (704 cases and 9,031 controls) and three late-onset case-control subgroups (1,467 cases and 19,156 controls).

Cases in the replication cohort were stratified into early-onset ($n = 375$, 2–11 years) and late-onset ($n = 977$, 21–81 years) subgroups as above. To compensate for sample size differences between the two subgroups, we used simple random sampling implemented in the SURVEYSELECT procedure in SAS version 9.4 to randomly assign two-thirds ($n = 1455$) of controls to the early-onset subgroup and one-third ($n = 727$) to the late-onset subgroup.

**Association analyses.** We performed separate GWAS in the early- and late-onset subgroups, with no overlap of subjects between the two subgroups. To control for population stratification, we performed Cochran-Mantel-Haenszel (CMH) analysis using GemTools-derived clusters from each GWAS as strata. For chromosome X variants, we assumed complete X-inactivation and similar effect sizes in males and females, with the effect of one risk allele in males equal to the effect of two risk alleles in females. Excluding variants within the extended MHC, the genomic inflation factors for the early-onset and late-onset subgroups were 1.046 and 1.064, respectively. For each variant that achieved genome-wide significant association ($P_{CMH} < 5.0 \times 10^{-8}$) in either or both age-of-onset subgroups, we tested whether the effect size was significantly different in the two subgroups using a $Z$ test[32], calculating $Z$ as the difference between the logarithms of the ORs, divided by the square root of the sum of the variances of the logarithms of the ORs, where $Z$ follows a normal distribution. Given 6,273 SNPs tested genome-wide, and with many SNPs in the MHC region in linkage disequilibrium, we considered $P_Z < 5.0 \times 10^{-5}$ as the significant effect size difference criterion.

For the replication study, we performed logistic regression analyses of rs145954018 and rs9271597 in the early- and late-onset subgroups.

For all logistic regression analyses using the GWAS or GWAS plus replication study data, we included significant eigenvectors for each GWAS as covariates to control for population stratification.

To test whether there are other variants representing the same signals as rs145954018 and rs927159 in the combined GWAS data, we performed logistic regression analysis for all MHC region variants conditional on rs145954018 or rs927159. If a tested variant and the conditional variant in the same model could not improve each other significantly (both with $P_{conditional} > 0.05$), both were considered to represent the same signal.

To compare dominant versus additive effect models for the rs145954018del-rs9271597A haplotype, we first fit a logistic regression model with the additive effect of the haplotype (coded as 0, 1, 2 for having 0, 1, or 2 copies) and an extra term for dominance effect (coded 1 for possessing exactly 1 copy of the haplotype and 0 otherwise; Supplementary Table 4). As a significant dominance effect was detected ($P = 1.06 \times 10^{-2}$), we tested complete versus incomplete dominance by

comparing the BIC of the model with complete dominance effect (coded as 0, 1, 1 for having 0, 1, or 2 copies of the haplotype) versus the model with both additive and dominance effects, and the model with complete dominance effect showed lower BIC, suggesting that a model in which the haplotype is completely dominant is a better description of the data. We used the combined data for the early-onset subgroup from the three GWAS and the replication study for the analysis.

To dissect the relative contributions of rs149554018, rs9271597, and the classical *HLA* alleles to vitiligo risk in the combined GWAS data, we compared logistic regression models of haplotypes defined by different combinations of rs145954018, rs9271597, *HLA-DRB1*13:01*, *HLA-DQA1*01:03*, and *HLA-DQB1*06:03*, and used BIC to select the best fit model.

To calculate the odds ratio of developing vitiligo at age of 5–9 years for individuals carrying the rs145954018del-rs9271597A haplotype versus not, we performed logistic regression analysis using cases with age-of-onset of 5 to 9 and Gem-tools[12] genetic-matched controls from the three GWAS and the replication study. A total of 464 cases and 10,720 controls were used for the analysis.

We compared frequencies of vitiligo cases with self-reported affliction with other, concomitant autoimmune diseases including type 1 diabetes, Grave's Disease, Hashimoto thyroiditis, Addison disease, systemic lupus erythematosus, pernicious anemia, and rheumatoid arthritis in early- and late-onset cases using Fisher's exact test.

To test association between non-vitiligo autoimmunity and *HLA-DRB1*13:01* and rs145954018, we coded vitiligo cases having at least one other vitiligo-associated autoimmune diseases (autoimmune thyroid disease, type 1 diabetes, pernicious anemia, rheumatoid arthritis, systemic lupus erythematosus, and Addison disease) as 1 and 0 otherwise, and used this phenotype as the responsible variable in logistic regression models, testing the additive effects of *HLA-DRB1*13:01* and rs145954018del both individually and together. 540 vitiligo cases from the combined GWAS having at least one other vitiligo-associated autoimmune diseases and 1,824 without were used in the analysis. For protein expression, we performed linear regression analysis to test separately the association between HLA-DQ and HLA-DR protein expression levels and the allele counts of rs145954018del, rs9271597A, and haplotype count of rs145954018del-rs9271597A.

**Estimation of heritability**. To estimate total vitiligo heritability in the separate early-onset and late-onset subgroups, we performed LD- and MAF-stratified GREML (GREML-LDMS)[33] analysis using the genome-wide imputed genotype data of the two subgroups, including the significant eigenvectors for each of the three GWAS as covariates in the analysis. After quality control procedures (imputation $r^2$ info > 0.6, MAF > 0.01, and Hardy–Weinberg equilibrium $P > 1 \times 10^{-5}$), a total 8,593,074 variants were retained for the analyses. To estimate the phenotypic variance explained by the previously identified non-MHC and MHC class I vitiligo susceptibility loci, we examined all variants within each non-MHC locus, defining region boundaries where the $P_{CMH}$-values gradually increased to 0.05, and we specified the MHC class I region as previously defined[34]. We then re-ran GREML-LDMS using the genome-wide imputed data excluding all variants within the region boundaries defined for non-MHC loci and the MHC class I region. The difference between the total phenotypic variance explained before and after removing variants at the non-MHC loci and MHC class I region was considered as the phenotypic variance explained by these loci. We calculated phenotypic variance explained by rs145954018 and/or rs9271597 as the McFadden's pseudo $r^2$ difference between the logistic regression model including the significant eigenvectors for each of the three GWAS plus either or both variants and the model including only the significant eigenvectors for each of the three GWAS.

**Analysis of MHC class II protein and mRNA**. We selected 46 available unrelated, healthy subjects ($n = 27$ male, $n = 19$ female) of non-Hispanic European ancestry with no known autoimmune diseases or current infections from the replication cohort of our previous GWAS[5] based on genotypes for rs145954018 and rs9271601 (genotypes: rs145954018: GT/GT, $n = 33$; del/GT, $n = 12$; del/del, $n = 1$; rs9271597: T/T, $n = 13$; T/A, $n = 20$; A/A, $n = 13$; inferred haplotypes: TG-T | TGT, $n = 13$; TG-T | TG-A, $n = 12$; TG-A | TG-A, $n = 8$; TG-T | del-A, $n = 8$; TG-A | del-A, $n = 4$; del-A | del-A, $n = 1$).To assay leukocyte surface HLA-DR and HLA-DQ proteins, peripheral venous blood was collected and flow cytometry was performed (Supplementary Fig. 4), using as antibodies PerCP-Cy5.5 anti-CD1c (BioLegend Cat# 331514, RRID AB_1227535); Brilliant Violet 785 anti-CD3 (BioLegend Cat# 317330; RRID AB_2563507); PE-Cy7 anti-CD11b (BioLegend Cat# 301322; RRID AB_830644); Brilliant Violet 421 anti-CD11c (BioLegend Cat# 301628; RRID AB_11203895); APC-Cy7 anti-CD14 (BioLegend Cat# 367108; RRID AB_2566710); Alexa Fluor 488 anti-CD19 (EBioscience Cat# 53-0199-42; RRID AB_1659677); APC anti-HLA-DR (BioLegend Cat# 307610; RRID AB_314688); PE anti-HLA-DQ (BioLegend Cat# 318106; RRID AB_604129) without dilution. Mononuclear cells were selected based on size and granularity and leukocytes were identified by CD45 staining. To exclude T and B cells, the CD3 and CD19 double negative population was selected and CD14 and CD11b double positive cells were used to identify monocytes. Mean fluorescence intensity (MFI) of HLA-DQ and HLA-DR expression on monocytes (CD3−, CD11b+, CD11c−, CD14+, CD19−), dendritic cells (CD3−, CD11b−, CD11c+, CD14−, CD19−), B lymphocytes (CD3−, CD11b−, CD14−, CD19+), and T lymphocytes (CD3+, CD11b−, CD11c−, CD14−, CD19−) was reported.

For RNA-seq analysis, three healthy subjects were selected from the protein expression study cohort above; one had phased rs145954018-rs9271597 genotypes TG-T | TG-T and two had phased genotypes TG-T | del-A. For each subject, peripheral venous blood was obtained and whole blood RNA-seq was performed. For allele-specific expression analysis, we first used seq2HLA[35] to perform HLA-typing using RNA-seq from each of the three subjects. We then aligned RNA-seq reads only to reference sequences for that subject's specific HLA type. The number of reads aligning to the most-likely *HLA-DRB1*, *HLA-DQA1*, and *HLA-DQB1* alleles in each subject were counted and presented as a ratio, reflecting the relative expression of each allele. Quantification of total *HLA-DRB1*, *HLA-DQA1*, and *HLA-DQB1* (RPKM) was provided by the standard output of seq2HLA.

**Reporting Summary**. Further information on experimental design is available in the Nature Research Reporting Summary linked to this article.

## Data availability

Case genotype and phenotype data for GWAS1, GWAS2, and GWAS3 subjects, and RNA-seq data for the present study, have been deposited in the Database of Genotypes and Phenotypes (dbGaP) under accession numbers phs000224.v1.p1, phs000224.v2.p1, phs000224.v3.p2, and phs000224.v4.p2). Vitiligo susceptibility and age-of-onset GWAS summary statistics have been deposited in the NHGRI-EBI Catalog of published genome-wide association studies. The source data underlying Figs 1, 2a and b, 3, 5a and b, and Supplementary Figs 1a and b, 2a-l, and 3a-d are provided as a Source Data file.

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

## Acknowledgements

This work was supported by grants R01AR056292, R01AR057212, R01AR065951, and P30AR057212 from the National Institutes of Health. Thanks to Katrina Diener and Theodore Shade in the University of Colorado Denver Genomics and Microarray Core for performing RNA-seq analysis and Ken Jones of the Bioinformatics Core for assistance with RNA-seq data analyses.

## Author contributions

All authors contributed extensively to the work reported here. R.A.S., S.A.S., B.E.P, and G.H.L.R. designed experiments, analyzed data, and wrote the manuscript; N.v.G., A.W., and K.E. recruited subjects; T.M.F., S.B., and P.N. performed experiments; Y.J. performed statistical analyses; and J.S. provided computational management.

## Additional information

**Competing interests:** The authors declare no competing interests.

