## [Peer Review File · Nature Communications]

Reviewers' Comments:

Reviewer #1:

Remarks to the Author:

I previously reviewed this manuscript for a different journal about 8 months ago. At that time, I thought the central claims were strong, interesting, and of broad relevance; I had some concerns with regard to the analysis that I recommended be addressed in a revised manuscript.

In the current submission, the authors have not implemented the specific type of analysis I suggested (treating age as a quantitative rather than categorical trait), but have extended considerably their findings and the implications, so much so that I no longer see a need to implement my earlier suggestion.

In brief, data from a multi-site GWAS for vitiligo is analyzed by age of onset. Based on an apparent bimodal distribution in the age of onset, the authors carry out separate analyses of "juvenile" age of onset and "adult" age of onset cases, and identify a single bp indel in an MHCII regulatory region that affects risk in the juvenile but not the adult age of onset group. The risk-associated allele of the indel is also associated with increased expression of two HLA-DQ genes, and a relative reduction in the frequency of other autoimmune conditions. The authors suggest that vitiligo may represent two groups of conditions whose underlying pathogenesis differs according to a single MHCII regulatory variant, age of onset, and likelihood of other autoimmune conditions.

Overall, I think the work is rigorous, important, and of significant interest to biomedical scientists across a wide range of disciplines. In many ways, the observations are reminiscent of the landmark paper from Mary-Claire King and colleagues in which stratification by age of onset for breast cancer revealed the existence of BRCA1.

I have several comments for the authors to consider:

1) The analysis of additivity vs. dominance is interesting, but I worry that the confidence intervals are large and there may not be enough power to distinguish among those alternatives. Given the postulated mechanism of action, and given the precedent for other complex human disease, true dominant action would be very unusual. The authors should consider these points, and perhaps back off the conclusion regarding dominance.

2) I do not completely understand the basis for carrying out allele-specific expression analysis, which makes reference to a manuscript published in 2012. My lack of understanding stems from an apparent inconsistency between the statements that: (1) causal variation in the region is regulatory rather than coding (a valid claim); and (2) coding variation has to be the basis for distinguishing allele-specific expression. I suspect that the allele-specific calls are based on synonymous or untranslated regions of the mRNA; however, examining and explaining this point would improve the presentation and accessibility of the work.

3) Besides susceptibility to other autoimmune diseases, it is important to know whether there are additional phenotypic differences (severity, distribution, anatomic site) that distinguish the two groups.

4) Is the indel derived or ancestral? What can the authors tell us about its evolution?

5) Likewise, it appears to me that the MAF of the indel is similar among most populations (0.03-0.05), except for populations of East Asian ancestry, where the MAF is much smaller. What might this mean

for the pathogenesis of vitiligo in populations of African and South Asian ancestry, where the condition is often more debilitating.

Reviewer #2:

Remarks to the Author:

In this paper the authors investigate the functional effects of MHC genetic variants associated with the autoimmune disease, vitiligo. They conclude that the genetic variation impacting on class II HLA expression is an important mechanism underlying the association. This concept has been suggested by several investigators in autoimmunity, but this paper has the potential to provide some important evidence to support this concept.

Major points

1. I was slightly unclear on the imputation strategy used. Were different GWA studies imputed using different methods, i.e. PBWT and IMPUTE? If so I think this has the potential to introduce significant bias. A more reliable plan would be to use genotype data from all GWAS and re-impute them together using the same method.
2. The authors should also clearly state whether the 'top' SNPs in each age-cohort were genotyped or imputed.
3. Were the key variants imputed or genotyped in the controls?
4. The authors data on distinct association signals based on a subset of the cohorts in which age of onset can be grouped is very interesting. Could the authors check whether there is any variation in the frequency of the risk two-variant haplotype with age in the controls?
5. The inclusion of DRB1*1301 in the association model makes only a small change to the BIC (Supp Table 8). Given that the HLA allele is imputed then erroneous imputation could account for the small change seen in the model fitting. Thus, I think to support the conclusion of the authors about the risk variants on this haplotype then some additional HLA typing is required. How many recombinants were observed between the risk haplotype and of DRB1*1301? Could the authors undertake targeted HLA typing on those subjects who carry alleles that are in strong LD with DRB1*1301?
6. Did the authors examine the relationship between genotype and HLA transcript abundance in the dataset presented in Figure 5 (cell surface HLA expression)?
7. It would be valuable to look in other ex vivo purified cell datasets cataloguing gene expression to determine whether the risk alleles described in this paper are associated with elevated expression of HLA class II.

Minor points

1. I think the authors overstate the case that MHC associations have been usually attributed to coding HLA variations in autoimmunity (Introduction – line 53). The mechanisms underlying the haplotypic associations tagged by HLA-DRB1*0301 and DRB1*15 that are present in several autoimmune diseases have not been clearly shown to be due to coding variants in the majority of cases.

Responses to Reviewers

Reviewer #1:

1) The analysis of additivity vs. dominance is interesting, but I worry that the confidence intervals are large and there may not be enough power to distinguish among those alternatives. Given the postulated mechanism of action, and given the precedent for other complex human disease, true dominant action would be very unusual. The authors should consider these points, and perhaps back off the conclusion regarding dominance.

As suggested by the reviewer, we have softened the language somewhat. Nevertheless, comparison of statistical models does suggest that a complete dominant model fits the data best. We have revised Supplementary Table 6 to make this clearer. A significant dominance term ($P = 1.06 \times 10^{-2}$) in a model with additive effect indicates that an additive effect alone is not sufficient. A complete dominant effect model had the lowest BIC and thus fits the data best.

*In fact, a dominant model would make reasonable biological sense. Cell surface HLA DQ is an obligate heterodimer comprised of one MHC class II alpha chain and one MHC class II beta chain. At some point, increased expression of beta (DQB1*06:03) chain likely saturates available alpha chain. That might be achieved by having one copy of the early-onset haplotype, in which case there would no additional biological effect of a second copy.*

2) I do not completely understand the basis for carrying out allele-specific expression analysis, which makes reference to a manuscript published in 2012. My lack of understanding stems from an apparent inconsistency between the statements that: (1) causal variation in the region is regulatory rather than coding (a valid claim); and (2) coding variation has to be the basis for distinguishing allele-specific expression. I suspect that the allele-specific calls are based on synonymous or untranslated regions of the mRNA; however, examining and explaining this point would improve the presentation and accessibility of the work.

We apologize for lack of clarity. For most genes, non-synonymous DNA sequence variation is uncommon and thus is not a good basis for distinguishing allele-specific expression. However, for most HLA genes non-synonymous DNA sequence variation is very common and therefore constitutes the standard basis for primary (4-digit) typing of HLA gene alleles. Accordingly, seq2HLA utilizes RNAseq data to assign standard 4-digit HLA allele calls based on non-synonymous variation. We have clarified this point in both the Results and in the Methods.

3) Besides susceptibility to other autoimmune diseases, it is important to know whether there are additional phenotypic differences (severity, distribution, anatomic site) that distinguish the two groups.

We appreciate this suggestion. In fact, we did consider body surface area quartile as well as acrofacial presentation; however, after adjusting for disease duration, there is no significant difference between the early-onset and late-onset case groups for either trait (P -value = 0.963 and 0.224, respectively). For other sub-phenotypic measures small numbers provide very limited statistical power. As the results of these analyses are negative, we don't feel they warrant presentation in the current paper.

4) Is the indel derived or ancestral? What can the authors tell us about its evolution?

As requested, we have added to the Discussion that the immediate rs145954018 ancestral allele is TG. The indel has not been observed in any species other than modern humans. In Denisova the sequence is TG with a common SNP [TA]; unfortunately, in Neanderthal there is as-yet no genome assembly in the MHC region. In other primates the sequence is TG in chimp, gorilla, bonobo, gibbon, proboscis monkey, snubnose monkey, Angolan colobus, crab-eating macaque, rhesus, sooty mangabey, green monkey, and drill; TA in orangutan, baboon, marmoset, and tarsier; CA in squirrel monkey and Ma's night monkey; AA in white-faced sapajou; and the sequence is unalignable in pig-tailed macaque, Sciater's lemur, black lemur, mouse lemur, and bushbaby. In other mammals alignments in the MHC are not reliable, though dog is assigned TA and armadillo AG. In other species no alignments are possible. With respect, we don't see an interesting evolutionary story that is relevant to the current manuscript.

5) Likewise, it appears to me that the MAF of the indel is similar among most populations (0.03-0.05), except for populations of East Asian ancestry, where the MAF is much smaller. What might this mean for the pathogenesis of vitiligo in populations of African and South Asian ancestry, where the condition is often more debilitating.

We agree that this is potentially very interesting. Accordingly, we have amended the Discussion to read, "Compared to the ancestral TG allele, the high-risk deletion allele is uncommon, with MAF 3 to 4% in most human populations (https://www.ncbi.nlm.nih.gov/projects/SNP/snp_ref.cgi?rs=145954018). Interestingly, its frequency is much lower, MAF about 0.5%, in east Asians, a population in which the prevalence of vitiligo is likewise much lower than in others²⁶⁻²⁸; whether the lower MAF of rs145954018 might in part account for the correspondingly lower prevalence of vitiligo remains unknown."

*We also agree with the reviewer that it will eventually be of great interest to compare the molecular pathogenesis of vitiligo across populations, especially given greater perceived "severity" of vitiligo in darker-skinned populations (not reflected in actual clinical differences) due to cultural stigmatization. Unfortunately, current knowledge of vitiligo genetics in non-EUR populations remains limited. Zhang and colleagues conducted GWAS of vitiligo in Chinese, and we conducted small GWAS of vitiligo in Japanese and the Indian-Pakistanis. The results show both similarities and differences, which as yet add up to no clear biological story. In the MHC, EUR has both class I (HLA-A*02:01) and class II associations; Japanese share that class I association but have no class II association; Chinese have MHC associations that remain quite unclear; and Indian-Pakistanis have no class I association and have a major class II association that is similar to yet different from the major EUR class II association. We are aware of no genetic data about vitiligo in any African-derived population. With respect, we believe such discussion is beyond the scope of the present paper.*

Reviewer #2:

Major points

1. I was slightly unclear on the imputation strategy used. Were different GWA studies imputed using different methods, i.e. PBWT and IMPUTE? If so I think this has the potential to introduce significant bias. A more reliable plan would be to use genotype data from all GWAS and re-impute them together using the same method.

We agree that our previous description of imputation was unclear. The three GWAS were performed using different genotyping arrays; therefore, it would be inappropriate to impute the

three GWAS together. As described previously, the data for each GWAS were imputed separately, each using the same method. We have substantially revised the description of imputation in the Methods to improve clarity, as follows:

Genome-wide genotyping, quality control procedures, and genome-wide imputation were described previously^{2,4,5}. In our previous genome-wide imputation of each of the three GWAS, we used SHAPEIT version2 (<http://www.shapeit.fr/>) to pre-phase genotypes to produce best-guess haplotypes, and then performed imputation with these estimated haplotypes using IMPUTE2 (https://mathgen.stats.ox.ac.uk/impute/impute_v2.html) and the 1000 Genomes Project phase I integrated variant set version 3 (March, 2012) as the reference panel. To increase genetic resolution in the current study, we used the Sanger Imputation Service (<https://imputation.sanger.ac.uk/>) to impute SNP genotypes using the Haplotype Reference Consortium (HRC) reference panel (release 1.1) (<http://www.haplotype-reference-consortium.org/>). For each GWAS, we used Eagle2³¹ to pre-phase genotypes to produce best-guess haplotypes and then performed imputation using PBWT³². SNPs with imputation INFO > 0.5, minor allele frequency ≥ 0.001 from the three GWAS combined, and without significant ($P < 1 \times 10^{-5}$) deviation from Hardy–Weinberg equilibrium were retained and combined with variants previously genotyped or imputed with IMPUTE2³³ from the previous GWAS⁵; for SNPs imputed by both IMPUTE2 and PBWT, the PBWT imputed genotypes were retained. We additionally imputed classical HLA alleles and amino acid polymorphisms for the three previous GWAS using SNP2HLA and the Type 1 Diabetes Genetics Consortium reference panel¹⁴. In total, 53,976 MHC variants and MHC classical alleles were tested in the final analysis. Primary genotype data for all three vitiligo GWAS are available for dbGaP (accession numbers phs000224.v1.p2, phs000224.v2.p2, phs000224.v3.p2).

2. The authors should also clearly state whether the ‘top’ SNPs in each age-cohort were genotyped or imputed.

Both rs145954018 and rs9271597 were imputed in cases and controls in the GWAS; as requested, we have added this statement to the Results. As we already described in the Results, both SNPs were genotyped in the replication study in both cases and controls.

3. Were the key variants imputed or genotyped in the controls?

See point 2, above

4. The authors data on distinct association signals based on a subset of the cohorts in which age of onset can be grouped is very interesting. Could the authors check whether there is any variation in the frequency of the risk two-variant haplotype with age in the controls?

With respect, we are puzzled by this request. The two-variant haplotype is associated with vitiligo age of onset in cases. Controls have no age of disease onset, and of course there is no relationship between genetics and chronological age of controls.

5. The inclusion of DRB1*1301 in the association model makes only a small change to the BIC (Supp Table 8). Given that the HLA allele is imputed then erroneous imputation could account for the small change seen in the model fitting. Thus, I think to support the conclusion of the authors about the risk variants on this haplotype then some additional HLA typing is required. How many recombinants were observed between the risk haplotype and of DRB1*1301? Could the authors undertake targeted HLA typing on those subjects who carry alleles that are in strong LD with DRB1*1301?

*The reviewer makes an excellent point, and we appreciate the prompt for deeper investigation. To make this clearer, we carried out additional analyses and have revised Supplementary Table 8 accordingly. Rather than providing just the overall P-value for each model, we now provide haplotype-specific P-values, which are most illuminating. As shown in the revised Supplementary Table 8, HLA-DRB1*13:01 occurs at frequency 0.039 on the high-risk rs149554018-rs9271597 del-A (“DA”) haplotype ($P = 3.81 \times 10^{-53}$), but it also occurs on the TG-A haplotype (“IA”) at a very similar frequency (0.030), but with far less significant P-value ($P = 4.45 \times 10^{-3}$). Therefore, HLA-DRB1*13:01 is much more common (frequency 0.069) than is haplotype DA but is far less significantly associated with vitiligo; therefore, HLA-DRB1*13:01 is unlikely to be the actual risk variant on this haplotype.*

Many tens of thousands of European-derived subjects have been analyzed to assemble extended high-resolution sequence-based HLA haplotypes as reference standards, with over 6.5 million additional subject haplotypes available at lower resolution. These data are available in the IPD-IMGT/HLA database, which serves as the global reference in immunogenetics and immunoinformatics. As such, we don't understand the request for additional HLA typing.

6. Did the authors examine the relationship between genotype and HLA transcript abundance in the dataset presented in Figure 5 (cell surface HLA expression)?

The purpose of the very limited RNAseq analyses summarized in Table 2 was to ascertain whether elevated HLA DQ protein levels associated with carriage of the early-onset/high-risk haplotype were correlated with elevated RNA from HLA-DQ alleles carried on that haplotype (that is, in cis). That indeed was the case. We have revised Figure 5 (showing the protein results) and Table 2 (showing the RNAseq studies) to color-code the three subjects studied by both protein analyses and RNAseq analyses, and we have revised the corresponding Figure and Table legends to indicate this.

7. It would be valuable to look in other ex vivo purified cell datasets cataloguing gene expression to determine whether the risk alleles described in this paper are associated with elevated expression of HLA class II.

With respect, despite considerable effort we can find no available relevant ex vivo purified cell datasets that would enable allele-specific HLA class II expression analysis as the reviewer suggests. HLA class II genes are expressed only by monocytes, dendritic cells, thymic epithelial cells, B cells (with major biological differences), and brain microglia. Allele-specific expression analyses of HLA class II genes requires individual-level RNAseq data (to assign allele-specific HLA expression) paired with either individual-level whole-genome sequence data (to derive SNP genotypes) or individual-level whole-genome imputed SNP genotype data from the same subjects. Furthermore, such data would be required from a large enough set of (hopefully normal) subjects to enable observation of individuals with the rs145954018 variant (MAF only ~0.035). Currently, the only somewhat-relevant available such datasets we can find are for unsorted whole blood (GTEx) and EBV-transformed lymphoblastoid cell lines (1KGP and GTEx). We have already assayed unsorted whole blood (this paper), and the biology of HLA class II expression by lymphoblastoid cell lines is so different from that of primary B cells it is unclear what would be the expected outcome.

Minor points

1. I think the authors overstate the case that MHC associations have been usually attributed to coding HLA variations in autoimmunity (Introduction – line 53). The mechanisms underlying the haplotypic associations tagged by HLA-DRB1*0301 and DRB1*15 that are present in several autoimmune diseases have not been clearly shown to be due to coding variants in the majority of cases.

As requested, we have softened this assertion at line 53 and throughout the manuscript. Nevertheless, it is certainly true that for most autoimmune diseases major attention in the MHC has long been focused on HLA coding variation, regardless of whether those ultimately prove to be the key drivers of disease.

Reviewers' Comments:

Reviewer #1:

Remarks to the Author:

In the revised manuscript, the authors have responded thoughtfully and clearly to my and the other reviewer's suggestions.

I think the work represents an important advance in our understanding of human complex disease, and I look forward to seeing it published.

Reviewer #2:

None